# Direction of Arrival Estimation and Highlighting Characteristics of Testing Wideband Echoes from Multiple Autonomous Underwater Vehicles

**DOI:** 10.3390/s23198318

**Published:** 2023-10-08

**Authors:** Xiaofeng Yin, Peizhen Zhang, Guangbo Zhou, Ziyi Feng

**Affiliations:** College of Electronic and Information Engineering, Guangdong Ocean University, Zhanjiang 524088, China; levines0769@163.com (X.Y.);

**Keywords:** wideband echo signal, autonomous underwater vehicles (AUVs), multiple targets, autofocusing, coherent signals

## Abstract

Multiple autonomous underwater vehicles (AUVs) have gradually become the trend in underwater operations. Identifying and detecting these new underwater multi-targets is difficult when studying underwater moving targets. A 28-element transducer is used to test the echo of multiple AUVs with different layouts in a lake. The characteristics of the wideband echo signals are studied. Under the condition that the direction of arrival (DOA) is not known, an autofocus coherent signal subspace (ACCSM) method is proposed. The focusing matrix is constructed based on the received data. The spatial spectrum of the array signal of multiple AUVs at different attitudes is calculated. The algorithm estimates the DOA of the echo signals to overcome the shortcomings of traditional wideband DOA estimation and improve its accuracy. The results show that the highlights are not only related to the number of AUVs, but are also modified by scale and attitude. The contribution of the microstructure of the target in the overall echo cannot be ignored. Different parts of the target affect the number of highlights, thus resulting in varying numbers of highlights at different attitude angle intervals. The results have significant implications for underwater multi-target recognition.

## 1. Introduction

AUVs, underwater unmanned vehicles, are widely utilized in geological exploration, including in mineral and energy resource surveys, underwater communication, anti-submarine warfare, and underwater search and rescue, and have gradually become a primary tool for underwater operations. To meet the increasingly complex task requirements, AUV technology is advancing towards swarm intelligence and increased autonomy [1,2,3]. AUVs not only function as equipment carriers, but also play a critical role in their own detection. Detecting AUVs underwater poses an exceptionally challenging task. Researching AUVs’ echo signal characteristics is of utmost importance.

However, the identification and detection of underwater targets heavily rely on underwater sonar arrays to primarily estimate the DOA of target azimuth information. Kazimierski [4] defined process noise and measurement models for underwater target tracking based on forward-looking sonar technology. This research made a significant contribution to maritime safety by improving the tracking accuracy and providing reference values for the covariance matrix in practical applications. Wawrzyniak [5] devised a method to precisely locate mechanically scanned sonar images by comparing them to a database of synthetic images created using bathymetric data. They explored various probability functions for image comparison, addressing the challenge of accurately georeferencing sonar images. The methods for DOA estimation have evolved from initial beamforming techniques to the extraction of feature information related to the target azimuth from the signal’s spectrum, which is often combined with machine learning for target recognition. Jiang [6,7,8] established a comprehensive set of multi-scale spectral features for underwater target recognition and effectively identified underwater targets using machine learning algorithms. Liu [9] trained a convolutional neural network model using covariance matrices from different orientations, effectively learning the differences between covariance matrices from different incident angles and accurately estimating the DOA.

Moreover, improving the receiving array can provide a higher estimation accuracy. Liu [10] utilized a large-space T-shaped array consisting of a generalized and symmetrically expanded co-prime array to estimate the DOA in two-dimensional space, eliminating angle ambiguity and providing a higher estimation accuracy. Liu [11] proposed a solution for DOA estimation by segmenting and extracting received pulse waves, which effectively reduces directional errors, as verified via experiments. Zhou [12] developed a RINA sensor array structure with increasing array degrees of freedom by employing hole-free co-array techniques. Based on this array, they introduced a C-CD algorithm to accurately estimate the azimuth of underwater target echoes. To detect multiple underwater targets, Moreno-Salinas [13,14] employed underwater AUVs as transducer-carrying platforms. By utilizing the distance information from received signals, they optimized the determinant of the Fisher information matrix, corresponding to the geometric configuration of the sensor network. This allowed the optimal positioning configuration of two underwater sensors to successfully identify two or three underwater targets. However, when dealing with more than three targets, the simultaneous localization and identification of all targets is only feasible with clear constraints on the target configuration. Zhang [15] proposed a DOA estimation method for two-dimensional power distribution. When the number of array elements in the base array is insufficient, this method yields accurate estimation results and high resolution, enabling high-precision DOA estimation with fewer base array elements.

Wideband signals offer several advantages over narrowband signals, such as that they carry a large amount of information in the target echo and exhibit weak reverberation background correlation, which is conducive to target detection, parameter estimation, and target feature extraction. Therefore, the use of wideband signals for underwater target detection has a distinct advantage. Jia [16] based their approach on the traditional planar element method and used Chebyshev polynomial interpolation to rapidly compute wideband echo characteristics of underwater targets. This approach is advantageous for calculating the complex echo characteristics of underwater targets.

Currently, research on the echo characteristics of AUVs primarily remains in the simulation stage, and our understanding of wideband echo signals of multiple underwater AUVs is not yet mature.

In the pursuit of underwater multi-target detection and identification, it is essential to understand the mechanisms of echo generation and the contributions of different targets within the overall acoustic scattering field. This paper explores essential techniques for characterizing multi-target properties, encompassing DOA estimation and the feature extraction of targets’ echo highlights. To assess the effectiveness of these methods, we conducted active sonar echo tests with multiple AUVs under several conditions in a lake. Due to the low accuracy of the incoherent signal subspace method (ISSM) and coherent signal subspace method (CSSM) algorithms in estimating the DOA, an autofocusing coherent signal subspace method (ACCSM) algorithm was proposed. This can be significantly impacted by previous angle estimation errors. ACCSM effectively handles coherent signal sources and avoids the prior angle estimation of the signal’s DOA, thereby enhancing the accuracy of DOA estimation. The results obtained through time–frequency spectrum and spatial spectrum estimations accurately determine the number of echo highlights and azimuth associated with the targets. The statistical results show that when the attitude angle has a range of −90–90°, there are different numbers of highlights at different attitude angle intervals.

## 2. Direction Estimation of Wideband Signal

For far-field narrowband signals, the envelope of the signal changes slowly within the time difference between arrivals at adjacent array elements. It can be approximated that a phase difference only exists in the signals received by adjacent array elements within this time difference. However, for wideband signals, the bandwidth relative to the center frequency cannot be ignored, and the direction vector of the array is not only related to the structure of the array, but also depends on the frequency of the signal. Therefore, narrowband DOA estimation algorithms are no longer applicable to wideband signals. Currently, there are many problems that need to be solved for algorithms of wideband DOA estimations, and there are two main types of mature algorithms. The first type is the ISSM, which is an extension of the multiple signal classification (MUSIC) [17] algorithm in the frequency domain. It has a simple principle but is associated with significant limitations. The second type is the CSSM, which can effectively deal with coherent sources, but requires prior knowledge of the source arrival angles.

### 2.1. Wideband Non-Coherent Signal Subspace Algorithm

The primary principle of ISSM is to decompose the incoming wideband signal into narrowband signals characterized by non-overlapping frequency bands. By analyzing the information within these narrowband components, it is possible to estimate the DOA of the wideband signal.

#### 2.1.1. Wideband Signal Model

Assuming an *M*-element linear array, there are *K*(*K* ≤ *M*) mutually incoherent wideband signals *S*_1_(*t*), *S*_2_(*t*), …, *S_K_*(*t*) from *K* sources impinging on the array transducer at angles *θ*_1_, *θ*_2_, …, *θ_K_*. Then, the signal received by the *m*-th array element is given by:(1)$$x\left(t\right)=\sum_{k=1}^{K}S_{k}\left(t-\tau_{m}\left(\theta_{k}\right)\right)+\eta_{m}\left(t\right),m=1,2,\ldots ,M$$

In Equation (1), *τ*_m_(*θ_k_*) is the phase delay associated with the transmission signal from the *k*-th source to the *m*-th array element, and *η_m_*(*t*) represents the additive noise at the *m*-th array element.

#### 2.1.2. Wideband Incoherent Signal Subspace Method

ISSM involves dividing the received wideband signal into *L* segments, and each segment is subjected to *P*-point Discrete Fourier Transform (DFT) to obtain *P* non-overlapping narrowband output data. The output data for the *l*-th segment at frequency *f_i_* can be expressed as follows:(2)$$X_{l}\left(i\right)=A\left(f_{i}\right)S_{l}\left(i\right)+N_{l}\left(i\right);i=1,2,\ldots ,P;l=1,2,\ldots ,L$$
(3)$$f_{i}=\frac{i}{P}f_{s}$$
where *X_l_*(*i*), *S_l_*(*i*), and *N_l_*(*i*) represent the DFT of the received data, the transmission signal, and the noise at frequency *f_i_*, respectively. Here, *f_s_* denotes the sampling frequency, and *A*(*f_i_*) represents the directional vector of the receiving array, which is expressed as follows:(4)$$A(f_{i})=[a_{\theta_{1}}(f_{i}),a_{\theta_{2}}(f_{i}),\ldots ,a_{\theta_{K}}(f_{i})]$$
(5)$$a_{\theta_{k}}(f_{i})=[e^{-j2\pi f_{i}\tau_{1}(\theta_{k})},e^{-j2\pi f_{i}\tau_{2}(\theta_{k})},\ldots ,e^{-j2\pi f_{i}\tau_{M}(\theta_{k})}{]}^{T}$$

In Equation (5), the superscript “T” represents the transpose operation for vectors or matrices. Assuming that the incoming signals from different sources are mutually incoherent, the covariance matrix of the received signals at frequency *f_i_* can be expressed as follows:(6)$$R_{x}(f_{i})=\frac{1}{L}\sum_{l=1}^{L}X_{l}(f_{i})X_{l}^{H}(f_{i})$$

In Equation (6), the superscript “H” represents the complex conjugate transpose operation for vectors or matrices. By applying a narrowband high-resolution algorithm such as MUSIC to the covariance matrix at frequency *f_i_*, the DOA estimation for that frequency can be obtained. By summing and averaging the DOA estimates of all the frequencies, the DOA estimation for the wideband signal can be obtained.

The ISSM method uses a frequency–domain transformation to decompose the signal into narrowband data, each corresponding to different sub-bands within the overall bandwidth. Subsequently, narrowband high-resolution methods are applied to process the data in each sub-band, which yields multiple sets of spectrum estimation values. These estimates are then weighted and processed to determine the DOA.

### 2.2. Coherent Signal Subspace Method with Autofocus

#### 2.2.1. Coherent Signal Subspace Method

Due to the lack of coherence handling capability in the ISSM algorithm, it cannot directly handle the DOA estimation of coherent signals. Therefore, when facing coherent signals from multiple targets, it is necessary to reduce or remove the coherence of the signals. To solve this problem, the CSSM is used to process wideband coherent signals. The CSSM method focuses the signal subspaces of each frequency onto the signal subspace at a specific reference frequency. The signal subspaces corresponding to all the frequencies are then estimated using narrowband high-resolution estimation. By averaging the estimated results, the DOA of the signal can be estimated.

For each frequency *f_i_*, a corresponding focus transformation matrix *T*(*f_i_*) is constructed. By multiplying the direction vector *A*(*f_i_*) by *T*(*f_i_*), the signal subspace of *f_i_* is focused onto the reference frequency *f*_0_, resulting in *A*(*f*_0_).
(7)$$T(f_{i})A(f_{i})=A(f_{0})$$

The focus transformation matrix *T*(*f_i_*) is obtained using the following equation:(8)$$T(f_{i})=V(f_{i})U^{H}(f_{i})$$
where *U*(*f_i_*) and *V*(*f_i_*) represent the left and right singular vectors of *A*(*f*_0_)*A*^H^(*f_i_*), respectively. By constructing the focus transformation matrix at different frequencies, one can obtain the corresponding narrowband data at each frequency:(9)$$Y_{l}(i)=T(f_{i})X_{l}(i)=T(f_{i})A(f_{i})S_{l}(i)+T(f_{i})N_{l}(i)=A(f_{0})S_{l}(i)+T(f_{i})N_{l}(i)$$

The covariance matrix at the corresponding frequency is given by:(10)$$R_{y}(f_{i})=\frac{1}{L}\sum_{l=1}^{L}Y_{l}(f_{i})Y_{l}^{H}(f_{i})$$

Similar to the ISSM, the signal subspace at each frequency is utilized for DOA estimation using the narrowband high-resolution method. By summing and averaging all the results, the DOA estimation of the wideband signal can be conducted.

The focusing matrix in the CSSM method is obtained as the singular value decomposition of *A*(*f*_0_)*A*^H^(*f_i_*). Therefore, it requires the pre-estimation of the DOA of the signal to obtain *A*(*f_i_*) at each frequency. However, this estimation is unpredictable in practical testing. Meanwhile, the performance of the CSSM is highly sensitive to errors in the pre-estimated angles, leading to biased final estimates. To overcome this limitation, an improved approach to ACSSM is proposed.

#### 2.2.2. Autofocus Coherent Signal Subspace Method

In the ACSSM, the covariance matrices at frequencies *f_i_* and the reference frequency *f*_0_ are computed using Equation (6), respectively. Subsequently, eigenvalue decomposition is applied to these covariance matrices to obtain the corresponding eigenvectors *G*(*f_i_*) and *G*(*f*_0_). The focusing matrix of ACSSM is given by:(11)$$T_{\mathrm{auto}}(f_{i})=\frac{1}{\sqrt{P}}G(f_{0})G^{H}(f_{i})$$

Narrowband data at the corresponding frequency are obtained as follows:(12)$$Z_{l}(i)=T_{\mathrm{auto}}(f_{i})X_{l}(i)=T_{\mathrm{auto}}(f_{i})A(f_{i})S_{l}(i)+T_{\mathrm{auto}}(f_{i})N_{l}(i)=A(f_{0})S_{l}(i)+T_{\mathrm{auto}}(f_{i})N_{l}(i)$$

The covariance matrix at the corresponding frequency is given by:(13)$$R_{Z}(f_{i})=\frac{1}{L}\sum_{l=1}^{L}Z_{l}(f_{i})Z_{l}^{H}(f_{i})$$

The narrowband processing method is applied to the covariance matrix at different frequencies to obtain the DOA results for wideband signals. Utilizing the ACSSM, there is no need for prior knowledge of the estimated angles, as the method relies entirely on the received data to construct the focusing matrix.

### 2.3. Simulation Analysis

Assuming a uniform linear array with eight elements, the incident angles of the sources in space relative to the array are *θ*_1_ = −30°, *θ*_2_ = −10°, and *θ*_3_ = 20°. The frequency of the impinging signals ranges from 500 to 1700 Hz, and the signal-to-noise ratio (SNR) is 10 dB. The sensor spacing is set to a half-wavelength corresponding to the highest frequency. The samples of the impinging signal are divided into *L* = 10 segments, each containing *P* = 256 samples. In each segment, 256 samples are transformed into the frequency domain using a 256-point DFT. DOA estimation is performed using three different algorithms: (i) ISSM; (ii) CSSM (with initial DOA estimates of −30°, −10°, and 20°); and (iii) ACSSM. Using the ACCSM algorithm, 1100 Hz is used as the reference frequency *f*_0_ and the focus matrix at different frequencies was constructed using Equation (11). Equation (12) is used to obtain narrowband data at different frequencies, facilitating the estimation of DOA. As shown in Figure 1, it can be observed that the ISSM algorithm and CSSM algorithm only exhibit significant peaks at −30° and −10°, while the peak at 20° is not prominent. On the other hand, the ACSSM algorithm shows peaks in three directions, indicating accurate DOA estimation results.

## 3. Experiment

### 3.1. Targets and Experimental Setup

During the experiment conducted in a lake to study the wideband echo characteristic of multi-AUVs, a formation of three AUVs was used. The experiment employs a monostatic transducer with 28-element for both transmitting and receiving signals. The transmitted signal frequency spans from 100 kHz to 200 kHz. Three targets of varying scale are utilized in the experiment, and images are shown in Figure 2.

AUV #1: This is the largest one, with a length of 2.3 m and a diameter of 0.2 m. Its main structure included a hemispherical head, a cylindrical body, a tail with propellers and four sponsons, and a top control cabin.

AUV #2: With a total length of 1.2 m and a diameter of 0.12 m, its components include a hemispherical head, a cylindrical body, four sponsons, a tail with a propeller, and two cylindrical control cabins on top.

AUV #3: This AUV measures 0.9 m in length and has a diameter of 0.12 m. It features a cylindrical body, a hemispherical head, and a tail with a propeller.

During the testing process, with the aim of enabling underwater communication between multiple AUVs and the formation control of AUVs [3], the three targets were arranged in two different formations for the analysis of echo characteristics: the triangular formation and the diagonal formation. For AUV #1, a 360° rotation was performed to determine the DOA of the target at different attitude angles, allowing the comprehensive analysis of echo characteristics. On the other hand, AUV #2 and AUV #3 maintained fixed positions throughout the experiment. During the test, all three targets were positioned at a depth of 9 m. The layouts of the formations, from a top-down perspective, are illustrated in Figure 3.

### 3.2. TimeFrequency Characteristics of Echo

The time domain echoes for the triangular and diagonal formations are shown in Figure 4a and Figure 5a, respectively. In addition, the azimuth angles of AUV #1 are shown in the figure, where 0° represents the vertical azimuth of the transducer with respect to the axis of the cylindrical body, −90° corresponds to facing the hemispherical bow section, and 90° indicates facing the tail section with the propellers. The angle-frequency plots for a single AUV #1 in these two formations are shown in Figure 4b and Figure 5b, respectively. In addition, Figure 4c and Figure 5c show the angle–frequency plots of the three targets, providing an insight into their acoustic characteristics.

From the spectrum in Figure 4 and Figure 5, it can be observed that the intensity of the targets is correlated with both the target frequency and angle, resulting in alternating light and dark stripes. By comparing Figure 4b,c, as well as Figure 5b,c, it can be seen that as the number of targets increases, the overall scattered energy becomes stronger, and the number of resonance peaks becomes larger. Although the differences between the single-target and multi-target echo characteristics can be seen from the time domain and frequency domain analyses in Figure 4 and Figure 5, these analyses alone cannot provide a clear understanding of individual target echo characteristics or determine the contributions of different target echo highlights and intensities of the overall resonance characteristics.

### 3.3. DOA Estimation

The distance-angle spectra for the two-dimensional distribution of highlights are computed using the conventional beamforming method [18]. Beamforming achieves directional focusing, creating highlights in specific orientations. In the diagonal formation, the distance-angle spectrum for beamforming at different azimuth angles of AUV #1 (relative to the transducer’s incident angle) is shown in Figure 6a–c for the −20°, 0°, and 90° attitude angles, respectively. 

In the triangular formation, the distance-angle spectrum for the echo beamforming at different azimuth angles of AUV #1 (relative to the transducer’s incident angle) is shown in Figure 7a–c for the −45°,0°, and 45° attitude angles, respectively. 

From Figure 6 and Figure 7, it can be observed that the beamforming method provides the preliminary estimation of the approximate locations of the echoes, but the number of precise DOAs cannot be determined with certainty. To achieve high-resolution spatial angle spectrum estimation and gain a more accurate understanding of the DOA, the ACSSM is employed again. Based on Equations (11)–(13), a reference frequency of 150 KHz is used to construct focus transformation matrices at different frequencies, resulting in corresponding narrowband data for each frequency. Taking the diagonal formation as an example, the spatial angle spectrum estimation is conducted for the attitude angles of −20°, 0°, and 90°.

Indeed, from Figure 8, it is evident that the distribution of echo highlights changes with different target attitude angles. The utilization of the ACSSM provides a more accurate reflection of the spatial angle spectrum for the three targets.

In Figure 8a, when the head of AUV #1 is rotated towards the transducer, the spatial spectrum of the multi-target echo signals shows four peaks. Among these peaks, the black and purple arrows represent the echo highlights received from #2 and #3, respectively, corresponding to the entire cylindrical body in the far field in the transverse orientation. The presence of strong incoming echoes around −5° and 8° is noted. The red arrows represent two echo highlights from #1’s head and the top control cabin. The incoming signals from directions −2° and 3° are well distinguished in this case.

In Figure 8b, the spatial spectrum exhibits five peaks, with #1 slightly deviating from the transverse orientation at a small angle. AUVs #2 and #3 remain relatively stationary, resulting in unchanged directions for their respective echo highlights. For AUV #1, the echo highlights from the head, top control cabin, and tail are clearly distinguishable, appearing at azimuth angles of approximately −2°, 1°, and 3°, respectively.

In Figure 8c, there are three peaks in the spatial spectrum, and AUV #1’s tail is rotated towards the transducer. Each target exhibits only one angle with a highlight. In this case, the tail of #1 plays a significant role in generating the echo highlight.

The azimuth angle estimation results of the echo signals for all the target attitudes in the experiment are presented in Figure 9a. This figure shows the DOA estimation results for all the spatial spectra. The histogram in Figure 9b illustrates the statistical distribution of peak angles derived from the spatial spectrum for all the target attitudes.

Figure 9 provides valuable insights into the distribution of echo highlights for the three targets in different orientations. Here are the key observations:

AUVs #2 and #3 (small AUVs) are represented by the green dots on the left and right sides in Figure 8a, respectively. The DOA distributions are observed at approximately −5° and 8°. This indicates that these two targets, being relatively smaller in scale and maintaining relatively stationary positions, only exhibit a significant echo highlight at these specific azimuth angles, while the other azimuth angles show less prominent distributions.

On the other hand, AUV #1, being larger in scale, exhibits variations in the number of highlights at different attitude angles. Analyzing the number of highlights provides valuable characteristic information about the target’s azimuth and attitude. The echo highlights for AUV #1 can be categorized into three regions for characterization.

(1)A single highlight region: In the range of attitude angles from −90° to −46°, AUV #1′s head is facing the transducer, and the main contribution to the echo comes from the head. Consequently, a single prominent echo highlight is observed in this region, with the primary azimuth angle measuring approximately 1°. Similarly, when the AUV is at an attitude angle from 43° to 90°, the tail faces the transducer, and only one echo highlight is observed due to the influence of the tail.(2)A double highlight region: In the range of attitude angles from −46° to −9°, as the target gradually rotates transversely, there are two main echo highlights observed at azimuth angles of approximately 2° and 3°. These echo highlights are primarily contributed by the head and the top control cabin. On the other hand, when the AUV is at an attitude angle from 2° to 40°, the target rotates from the transverse position towards the tail. As a result, two echo highlights become apparent, which are primarily influenced by the top control cabin and the tail section.(3)A triple highlight region: Within the attitude angles ranging from −9° to 3°, the number of prominent echo highlights increases to three. This phenomenon occurs as the target approaches a transverse orientation. The strong echo highlights mainly originate from the head, the cylindrical body, and the tail section. The probability distribution of azimuth angles exhibits peak values at approximately 1°, 2°, and 3°.

## 4. Conclusions

In this study, we conducted experiments on the wideband echo signals of three AUVs in a lake at different angles of attitude. The conclusions are as follows:

The received signals in the time domain and frequency domain alone could not provide a comprehensive understanding of the echo characteristics of different parts of the targets at different attitude angles. Furthermore, traditional beamforming methods exhibit limited accuracy in estimating the DOA of the targets. The ACCSM algorithm achieves higher accuracy in DOA estimation compared to those of the ISSM and CCSM algorithms, as demonstrated through simulation-based comparisons. Therefore, in this study, we utilized the ACCSM method to obtain detailed information about the echo highlights of the targets. This method improved the accuracy of wideband signal direction estimation by overcoming the need for prior angle estimation of the received signals in traditional wideband signal subspace methods.

In regard to the estimation of the DOA based on experimental data, the statistical analysis revealed the following findings: during the entire study of the echo characteristics, the echo highlights are not only related to the number of targets, but also to the azimuth and scale of the targets, which are important factors that cannot be ignored. The experimental results showed that the two smaller AUVs, due to their smaller scale and maintenance of a transverse orientation, exhibit only one prominent echo highlight under the influence of their cylindrical bodies.

On the other hand, when the targets are relatively larger in scale, their microscopic structures contribute significantly to the overall echo characteristics. This results in distinct echo highlight patterns for AUV #1 at different attitude angles. When the attitude angle is between −90° and −46° or 43° and 90°, the AUV’s head or tail faces the transducer, leading to a single prominent echo highlight. When the attitude angle is between −46° and −9° or 2° and 40, the AUV transitions from the transverse to head or tail orientation, resulting in the presence of two distinct echo highlights. When the attitude angle is between −9° and 3°, it is characterized by the appearance of three prominent highlights when the AUV approaches a transverse orientation.

It can be anticipated that when the sonar’s emitted signal frequency is higher, the microscopic structures of smaller-scale AUVs will also impact the overall scattering sound field. These analysis results are crucial for determining the presence of multiple underwater targets and their attitude angles within certain tracking angle ranges.

## Figures and Tables

**Figure 1 sensors-23-08318-f001:**
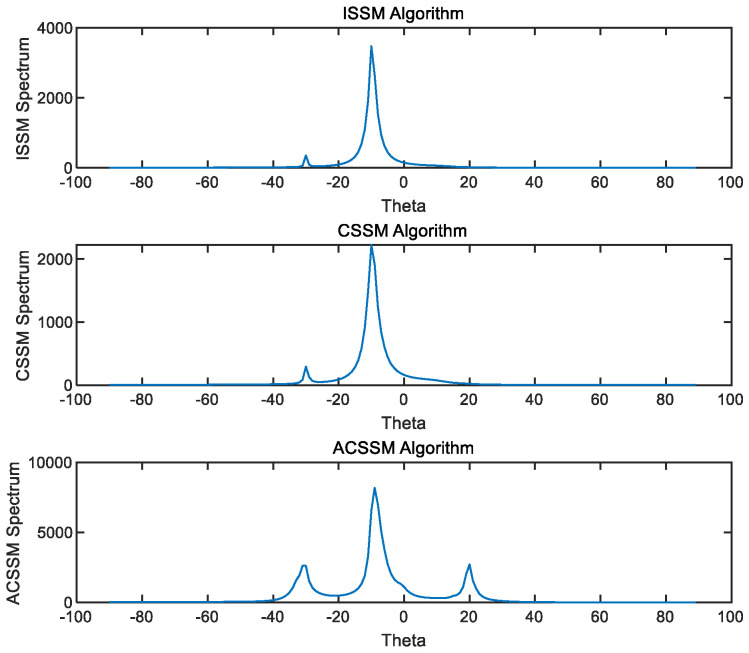
DOA estimation results of coherent signals.

**Figure 2 sensors-23-08318-f002:**
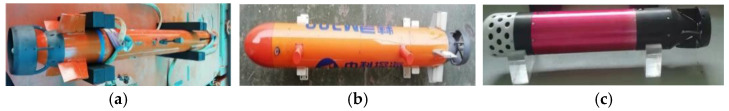
The pictures of AUVs. (**a**) AUV #1; (**b**) AUV #2; (**c**) AUV #3.

**Figure 3 sensors-23-08318-f003:**
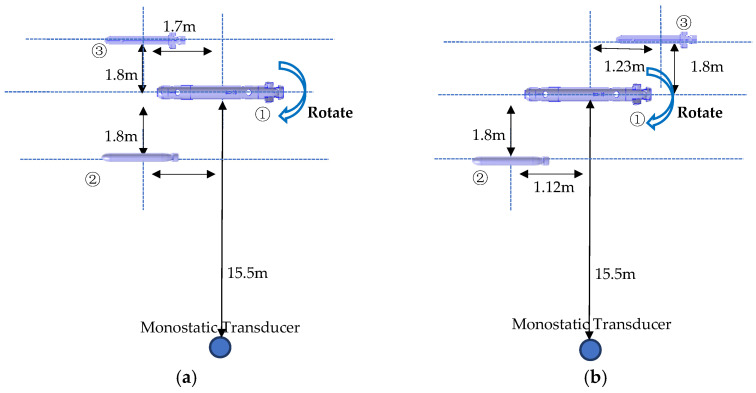
Layout of the experiment. (**a**) Triangular formation. (**b**) Diagonal formation.

**Figure 4 sensors-23-08318-f004:**
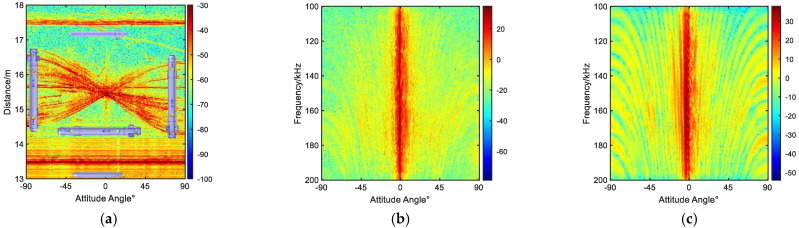
Diagonal formation. (**a**) Distance-angle time domain spectrum. (**b**) Target AUV #1 spectrum. (**c**) Frequency-angle spectrum of three targets.

**Figure 5 sensors-23-08318-f005:**
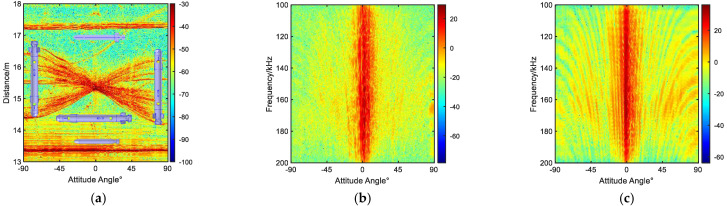
Triangular formation. (**a**) Distance-angle time domain spectrum. (**b**) Target AUV #1 spectrum. (**c**) Frequency-angle spectrum of three targets.

**Figure 6 sensors-23-08318-f006:**
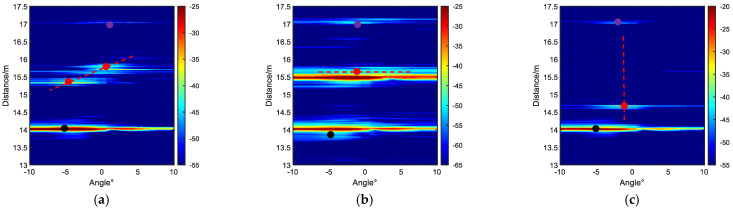
Distance-angle spectra for the two-dimensional distribution of highlights at different attitude angles. (**a**) −20°; (**b**) 0°; (**c**) 90°. (The red dashed lines represent the attitude status of AUV #1, the black dots and purple dots represent the echo highlights of AUV #2 and AUV #3, respectively).

**Figure 7 sensors-23-08318-f007:**
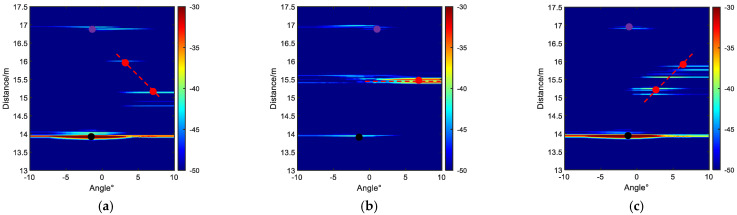
Distance–angle spectra for the two-dimensional distribution of highlights at different attitude angles. (**a**) −45°; (**b**) 0°; (**c**) 45°. (The red dashed lines represent the attitude status of AUV #1, the black dots and purple dots represent the echo highlights of AUV #2 and AUV #3, respectively).

**Figure 8 sensors-23-08318-f008:**
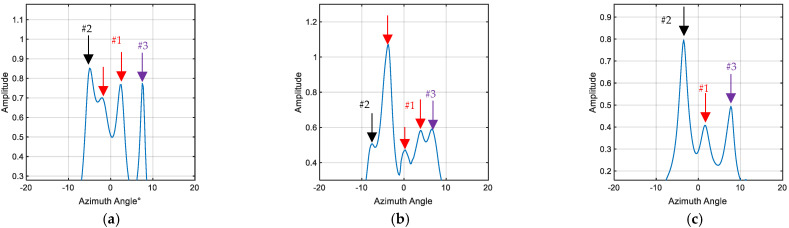
Spatial spectrum estimation of diagonal formation at different attitudes. (**a**) −20°; (**b**) 0°; (**c**) 90°. (The red arrows represent echo highlights from AUV#1, the black and purple arrows represent the echo highlights received from #2 and #3).

**Figure 9 sensors-23-08318-f009:**
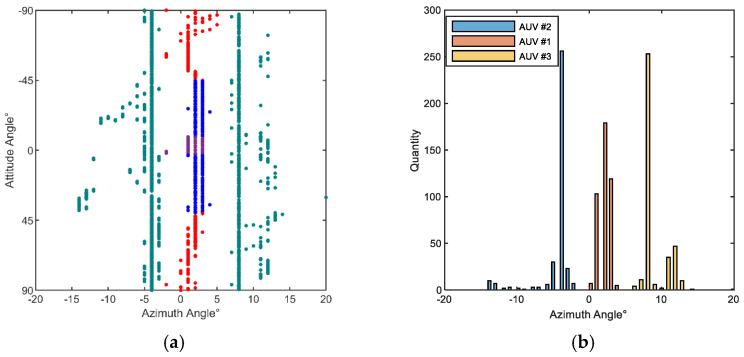
DOA and highlights distribution statistics. (**a**) The distribution of echo highlights. (**b**) Statistical distribution of highlights.

## Data Availability

The data that support the findings of this study are available from the corresponding author and first author upon reasonable request.

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
