# Peer review of "Direction of Arrival Estimation and Highlighting Characteristics of Testing Wideband Echoes from Multiple Autonomous Underwater Vehicles"

_sensors, 2023, doi:10.3390/s23198318_

Round 1
Reviewer 1 Report
1. The paper presents an experimental test on the wideband signal characteristics of multi-AUVs based on the auto-focus coherent signal subspace method (ACSSM). The experiment measured the distance and azimuth of three AUVs in two different formation setups, where one of the AUVs rotated 360o during the investigation. It is concluded that the number of AUVs and the rotation/attitude variation affected the measurement of distance and azimuth from the wideband transducer.
2. While the topic is interesting, the paper must be presented more systematically. Therefore, it requires significant revision in the writing and the experiment itself. There is no statement of the objective of the research, and the problem of the ISSM method that the author seems to want to solve using autofocus CSSM needs to be emphasized clearly.
3. The experiment setup focuses on the formation, but the conclusion is about how AUV's number and attitude affect the measurements. Why not set the experiment with a different number of AUVs? If formation is essential, it needs to be analyzed and concluded. And why use the triangular and diagonal formation in the first place? Is it related to how you want to prove that ACSSM is the best method compared to ISSM and CSSM? This experiment setup is separate from any objective and reasoning that should been set up in the introduction.
4. The methodology and reasoning could be more straightforward. ISSM needs to be appropriately introduced. Why is the method used? Is it the standard method? Is it the only method? The paper skips the method explanation, including the relation with the CSSM that was previously mentioned in the introduction and is the method that is used.
5. The beamforming method is not explained nor referenced.
6. Why do you use strange angle values in Figures 5 and 6 and their explanation? Why not simply use 90o, 45o and 0o?
7. Fig. 5 and 6 are supposed to show the distance and azimuth angle, but it needs to be clarified since the whole AUV is pictured. It will be much clearer if you put a centroid/point representing the vehicle, showing the resulting distance and azimuth (and the deviations?).
8. One of the conclusions is that ACSSM is better than CSSM and ISSM, but there is no comparative work, proof, or reference to back this conclusion up. Also, the comparison needs to be quantitative.
The English is fine grammatically, but there are many mistakes in the writing choice, such as:
1. There are two "underwater" in the title.
2. In the first paragraph, the sentence "an important detected underwater target in its own right" is hard to follow. What does that mean? Maybe use a more straightforward sentence like " It is important to detect them?"
3. Paragraph two should be split on the "However."
4. DOA abbreviation is introduced before explaining what it is abbreviated for. The paper also uses two terminology: "DOA" and "incoming wave direction." I suggest you use one terminology.
5. Some of the paragraph is not yet right-justified
6. Experiment and Test are synonyms. "Experimental Test" (Section 3) is redundant.
7. AUV #1, not AUV1#.
Author Response
Please see the attachment。

Reviewer 2 Report
The authors apply autofocus coherent signal subspace method (ACSSM) for detection and mainly characterizing of multiple autonomous underwater vehicles through analysis of wideband echo signals. This study focuses on the experimental side by analyzing echoes received from three AUVs in a lake under various attitude angles. The paper is well organized and clearly written. I have a number of minor editing suggestions for better clarity:
Line 13 – recommend changing " gestures" to attitudes or aspects.
Line 17 – recommend changing "bright spots" to highlights
Line 29 – "…evolving towards clustering….", maybe better using swarms instead of clustering?
Line 32 – 33 – Please rephrase correctly "it is urgent research on the characteristics of AUVs’ echo signals."
In addition, since the focus is the sea experiment and is its results and not the theoretical background, I suggest that this point will be mentioned clearly, maybe even in the title.
See comments and suggestions for the authors
Reviewer 3 Report
1. Identification and detection of underwater targets especially by sonar is one of the leading scientific problems of recent years. Both in civilian and military applications. In this paper an experimental test on the wideband signal characteristics of multi-AUVs was presented.
2. General remarks
a. Please use the language of a scientific research report without personal references: like “we” line 276.
b. The subject of underwater acoustic source recognition is commonly associated with sonar tracking, also using artificial neural networks. It is worth to compare algorithms used in acoustic source analysis idea to the solution proposed in the paper. Publications worth analyzing: doi: 10.1515/pomr-2017-0004 and doi: 10.3390/rs13051014.
c. However the article is very well written should be carefully edited. Only few remarks included below.
3. Specific remarks
a. Line 32 “underwater. it is urgent” should be “underwater. It is urgent”,
b. Line 196 “experiment(a)” should be “experiment (a)”,
c. Line 189, 227 , 240/241, 264 and others “.” should be on the end of sentence,
d. Line 241 missing “ “ after “)” – two times,
e. Line 324 “China(grant” should be “China (grant”,
f. The final conclusions are too general and only generally summarize the research presented in the article. I suggest expanding the conclusions with more detailed findings.
Round 2
Reviewer 1 Report
The new version is much clearer, and the review questions are well responded. The explanation is of the methods (ISSM, CSSM, and ACSSM) are added. The objectives and conclusions are now clear and well written. The new figures are understandable and illustrative. Still, there are some parts that need to be improved:
1. I still find it strange that the authors use the three strange angle values (22.3. 5.6, and 75.4 degree). Even stranger that the author actually can use any other number that they like (shown in the response). So why not just show a more standar number or integer number. Or at least give a sound reason of why those three values are used. Maybe because they show the most contrast?
2. The use of equation 11 - 13 (the ACSSM method) need to be referred adequately in the simulation / obtained result.
3. The numbers are not needed in the conclusion.
